# Tracking conformational transitions of the gonadotropin hormone receptors in a bilayer of (SDPC) poly-unsaturated lipids from all-atom molecular dynamics simulations

Eduardo Jardón-Valadez [1] *, Alfredo Ulloa-Aguirre [2,3] *

**1** Departamento de Recursos de la Tierra, Unidad Lerma, Universidad Autónoma Metropolitana, Lerma de Villada, Estado de México, Mexico, **2** Instituto Nacional de Ciencias Medicas y Nutrición "Salvador Zubiran". Mexico City, Mexico, **3** Red de Apoyo a la Investigación, Universidad Nacional Autónoma de México. Mexico City, Mexico

* h.jardon@correo.ler.uam.mx (EJ-V); aulloaa@unam.mx, alfredo.ulloaa@incmnsz.mx (AU-A)

## Abstract

Glycoprotein hormone receptors [thyrotropin (TSHR), luteinizing hormone/chorionic gonadotropin (LHCGR), and follicle stimulating hormone (FSHR) receptors] are rhodopsin-like G protein-coupled receptors. These receptors display common structural features including a prominent extracellular domain with leucine-rich repeats (LRR) stabilized by β-sheets and a long and flexible loop known as the hinge region (HR), and a transmembrane (TM) domain with seven α-helices interconnected by intra- and extracellular loops. Binding of the ligand to the LRR resembles a hand coupling transversally to the α- and β-subunits of the hormone, with the thumb being the HR. The structure of the FSH-FSHR complex suggests an activation mechanism in which Y335 at the HR binds into a pocket between the α- and β-chains of the hormone, leading to an adjustment of the extracellular loops. In this study, we performed molecular dynamics (MD) simulations to identify the conformational changes of the FSHR and LHCGR. We set up a FSHR structure as predicted by AlphaFold (AF-P23945); for the LHCGR structure we took the cryo-electron microscopy structure for the active state (PDB:7FII) as initial coordinates. Specifically, the flexibility of the HR domain and the correlated motions of the LRR and TM domain were analyzed. From the conformational changes of the LRR, TM domain, and HR we explored the conformational landscape by means of MD trajectories in all-atom approximation, including a membrane of polyunsaturated phospholipids. The distances and procedures here defined may be useful to propose reaction coordinates to describe diverse processes, such as the active-to-inactive transition, and to identify intermediaries suited for allosteric regulation and biased binding to cellular transducers in a selective activation strategy.

## Author summary

In the present study, we describe the results from a computational microscopy perspective (also known as molecular dynamics simulation) at the atomistic resolution for the two

**Data Availability Statement:** https://doi.org/10.5281/zenodo.10011977.

**Funding:** This study was supported by grant IN208323 from the Programa de Apoyo a Proyectos de Investigación e Innovación Tecnológica (PAPIIT), UNAM, Mexico (to A.U.-A).

**Competing interests:** The authors have declared that no competing interests exist.

gonadotropin hormone receptors, the follicle-stimulant hormone receptor and the luteinizing/chorionic gonadotropin hormone receptor, which are essential for reproduction in humans. Several dysfunctional mutations in these receptors, leading to reproductive failure, have been detected in the clinical arena. To better understand the process whereby these two receptors perform their signaling tasks, we assembled the active state of the receptor structures in a membrane bilayer of phospholipids with water molecules as solvent at both sides of the membrane, and without any agonist bound to the receptor. The systems included nearly 200 thousand atoms, each moving around at 300 kelvin and 1 bar given the interactions (attractive or repulsive forces) from each other. As the motion equations are solved in each time step (at femtoseconds time scale), the system evolved over time during hundreds of nanoseconds (millions of time steps) for three independent replicates. The receptor conformation displayed non-random motions due to the stability of specific structures in the complex molecular environment, including the hydrophobic membrane core, the bilayer interfaces, and the aqueous medium. From the analysis of simulation trajectories and structural changes of the receptors, we could identify the main conformational changes exhibited by each receptor explored in a model cellular environment. We discussed the role of the hinge region at the extracellular domain in triggering the receptor conformational changes, as well as differences in the dynamics between these two receptors in terms of the flexibility of the structures. Importantly, we proposed relative distances among the different receptor domains as parameters to characterize conformational intermediaries along a transition of states. Understanding of the signaling process in gonadotropin hormone receptors might be useful to explore new strategies for the modulation of the receptor functions, the bias of signaling pathways, or the selective binding of agonists.

## Introduction

G protein-coupled receptors (GPCRs) are a large and functionally diverse superfamily of plasma membrane receptors that respond to a widely variable endogenous and exogenous stimuli of diverse chemical structures, from photons, odorants, and ions to lipids, neurotransmitters, peptide hormones, and complex protein hormones. G protein-coupled receptors consist of a single polypeptide chain of variable length which traverse the lipid bilayer seven times forming characteristic transmembrane (TM) α-helices connected by alternating extracellular and intracellular loops (EL and IL, respectively), with an extracellular NH$_2$-terminus or ectodomain (ECD) and an intracellular COOH-terminus or Ctail [1,2]. These membrane receptors currently represent an important therapeutic target for several diseases in humans; in fact, ~30–40% of GPCRs of approved drugs target this family of membrane receptors [3,4].

The receptors for the glycoprotein hormones (GPHR)—follicle-stimulating hormone (FSH) receptor or follitropin receptor (FSHR), lutropin receptor or luteinizing hormone/chorionic gonadotropin (LH and CG, respectively) receptor (LHCGR), and thyrotropin receptor or thyroid stimulating hormone receptor (TSHR)—belong to a conserved subfamily of a GPCR family, the so-called Rhodopsin-like family, and more specifically, to the δ-group of this large class of GPCRs [1]. Structural features of the GPHRs include a large extracellular NH$_2$-terminus, where recognition and binding of their corresponding ligands (FSH, LH and CG, or TSH) occur. This ECD exhibits a central structural motif of imperfect leucine-rich repeats (LRR; 12 in the FSHR; 8–9 in the LHCGR; and 10 in the TSHR) [5–8], that is shared with other plasma membrane receptors involved in selectivity for ligand and specific protein-to-

protein interactions (Fig 1A)[9]. The carboxyl-terminal end of the ECD exhibits a domain critical for all GPHRs function: the so-called hinge region (HR); this particular region is involved in high affinity hormone binding, receptor activation and intramolecular signal transduction, and/or silencing of basal receptor activity in the absence of a ligand [10].

The gonadotropin receptors (FSHR and LHCGR) play an essential role in reproductive function. They regulate spermatogenesis and ovarian follicular maturation (FSHR) as well as steroidogenesis (testicular Leydig cells, ovarian follicle, and placenta) as well as ovulation (LHCGR) [7,11–13]. We have been interested in analyzing the effects of point mutations on

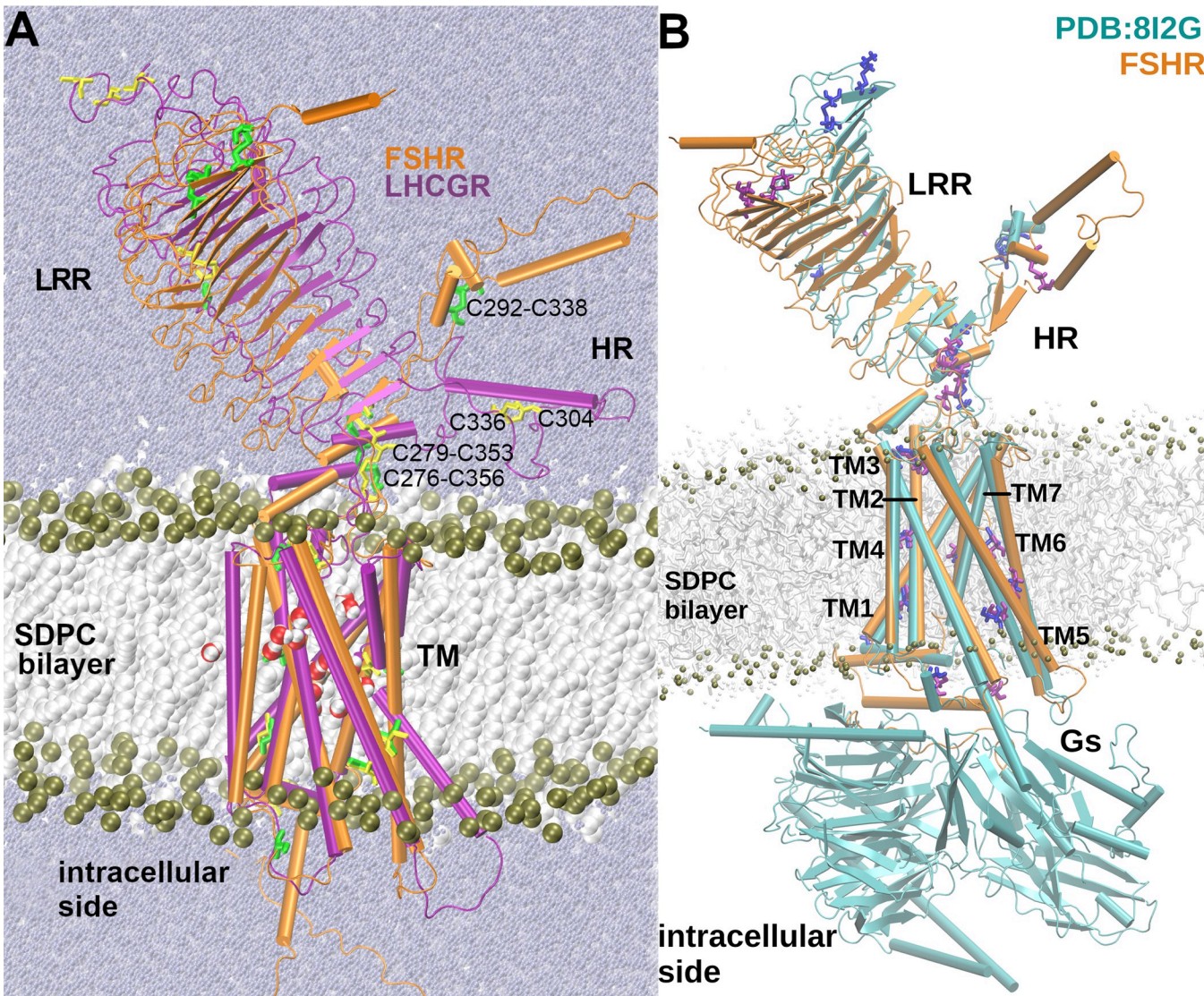

**Fig 1. A.** Simulation boxes for the FSHR and LHCGR (orange and lavender structures, respectively). The leucine rich region (LRR), hinge region (HR) and transmembrane (TM) domain are indicated. The SDPC bilayer is depicted with the lipid tails as white spheres, and the phosphorus atoms of the lipid heads as olive spheres. Solvent water molecules are depicted as small spheres in blue-gray. Side chains of cysteine residues are depicted in licorice, green for the FSHR and yellow for the LHCGR. Disulfide bonds are identified for C292-C338 and C276-C356 in the FSHR structure, and C279-C336 in that of the LHCGR. The bond C292-C338 at the HR of FSHR was not defined between C336 and C304, which are the homologous positions in the LHCGR. **B.** Structural alignment of the FSHR model (orange cartoons) and its corresponding cryo-EM structure (cyan cartoons) bound to its cognate agonist (not shown) and the Gs-protein. In the background as context, lipid tails are depicted as white sticks; phosphate atoms, as olive spheres; and cysteine atoms in licorice lavender and blue for FSHR and 8I2G structures, respectively. The conformation of both structures corresponds to the active conformation [21].

function and tridimensional structure of the FSHR [7,12,14]; in particular, on the understanding of the response of these receptors to their cognate agonists, departing from changes in the conformational dynamics related to the biological function in dysfunctional variants [14,15]. In principle, an extracellular signal is transmitted when the receptor is stabilized in an active conformation, allowing activation of the intracellular transducers coupled to the receptor. Nevertheless, the transition processes among different conformational states of the receptor triggered by the agonist are still incompletely understood [16,17]. For example, the FSHR may activate several intracellular signaling cascades in function of coupling to distinct pathways mediated by several kinases and/or β-arrestins, that is to say, the receptors are more than simple binary interruptors, but rather function as allosteric microprocessors that respond to the agonist stimulus with different affinities to distinct transducers: a given ligand may favor activation of a particular signal transducer and, in turn, the transducer may increase the affinity of the receptor to the ligand [18].

Departing from the elucidation of the LHCG receptor in the active and inactive state by cryo-electron microscopy (4.3 Å resolution), it is possible to identify the structural features of the GPHR subfamily [19]. The large extracellular ECD encompasses the LRR, similar to a boxing glove, with the thumb formed by the HR. The α-helix (P272-N280) and the loop P10 (F350-Y359) of the HR conform a communication interface between the LRR and the TM domain. For example, the α-helix Q425-T435 of the EL1 and P10 exhibit interactions in the inactivating S277I and activating E254K mutations, suggesting that the positions K354 and K605 are important regulators of the activation or inhibition of this receptor [20]. On the other hand, the C304-C336 disulfide bond was not observed in the crystal LHCGR structure, in contrast to the C292-C338 bond in the FSHR (Fig 1A). In studies previous to the report of the FSHR structure [21], our group proposed a model for this receptor considering only the TM domain (FSHR-TM) [15,22]. From the structural alignments of the LHCGR, FSHR and FSHR-TM, it was possible to identify that our FSHR-TM model corresponds to its active state (Figs 1 and S1). Among other findings derived from the trajectory analyses of molecular simulations of the FSHR-TM and D408Y and I423T inactivating mutants, it was possible to identify that helix 2 of the TM domain is an important communication hub for the propagation of intracellular signaling [14,15].

As a continuation of previous studies [14,15], in the present work we performed all-atom simulations for a FSHR model, but now including the corresponding LRR, HR and TM domain, as disclosed by the AlphaFold server (AF2; Fig 1). Examination of the conformational energy landscape was performed for a FSHR model and the structure of the LHCGR (PDB:7FII). Both GPCRs were relaxed in similar conditions through a simulation box that included water molecules as solvent, a polyunsaturated phospholipid membrane, and monovalent ions for charge balance. A difference to highlight between both structures is the presence of the disulfide bond C292-C338 at the HR of the FSHR, which is not observed in the equivalent position (C304-C336) in the LHCGR [19,20].

In the study of protein folding, it has been proposed the notion of a "energy funnel" that stretches down as the energy decreases [23,24]. When a protein is unfolded, without a well-defined structure, its possible conformations show a high variability that is represented by the top width of the funnel; as the molecular interactions favor the formation of contacts among residues, (eg. hydrogen bonds formation), the free energy of the protein decreases and the overall three dimensional shape of the protein (i.e. its stable set of conformations) is better defined, moving towards the lower portions of the funnel. When the protein reaches its native state, the overall conformation will move towards the end of the funnel exhibiting a minimal free energy. The progression of changes from the unfolded to the native (folded) state is rather a rough surface, with local minima and metastable states that should evolve to the more stable

conformations. In a given GPCR, as the FSHR, the surface of the conformational energy landscape may show more than one stable state that represents either the inactive state or distinct active, signaling states that may lead to full or biased signaling [25]. In this scenario, knowing the transition route to favor a particular signaling over other(s), represents, on one side, the understanding of the conformational changes occurring during the signal propagation process and, on the other, the possibility to control the selective signaling of a GPCR through different agonists, antagonists or allosteric modulators. The present study explores the energy landscape of the LRR-HR-TM domains in gonadotropin receptors to identify configurations compatible with the opening of their intracellular domains during activation.

From the computational point of view, the identification of critical sites for the allosteric regulation of protein function has been analyzed employing different approaches including machine learning, bioinformartics to detect allosteric sequences, site-directed mutagenesis, and molecular dynamics [26–28]. A widely employed technique to describe conformational changes of a protein is the principal component analysis (PC). Typically, the first PCs accumulate the largest variability with respect to a reference structure (*eg*. average structure). In the present study the exploration of the conformational surface of the GPCRs was performed through a combination of strategies, such as the use of the distances or angles between the LRR, HR and TM domain, and PC analysis. This study has the perspective of evaluating different coordinates of reaction to track the transition routes through the intermediates compatible with the conformational switches known to be involved in GPCR activation, namely the D/ERY ionic motif, the displacement of TM helix 6, and the rearrangement of TM helices 5 and 7 [2,29,30]. In addition, a trajectory of the LHCGR in 1-palmitoyl-2-oleoyl-sn-glycero-3-PC (POPC) was generated as the continuation of the initial processing in CHARMM-GUI for further comparisons.

## Results

### Detection of conformational changes from domain distances

Describing the conformational states for the glycoprotein hormone receptors obtained either by cryo-electron microscopy or computational models is relevant for the understanding of the mechanisms associated with receptor function. When the resolved structure is incomplete, modeling strategies emerge as an alternative approach to unveil conformational transitions at the atomic level. To convey the molecular complexity by which membrane receptors link an extracellular stimulus with an intracellular response, in Fig 1A we show the relaxed structures of the FSHR (AF-P23945) and LHCGR (PDB:7II) in a SDPC membrane environment, identifying the LRR, HR, and TM domains. Molecular dynamics (MD) simulations were performed to assess the dynamics of the receptors in a well-structured membrane, with a hydrophobic region, water-lipid interfaces, and bulk water according to the calculation of mass distribution across the bilayer normal (S2 and S3 Figs). Such membrane structure provided a stable molecular environment for preserving the integrity of the TM domains (S4 Fig). In addition, in Fig 1B a structural alignment of the FSHR model and the resolved structure for FSHR (PDB:8I2G) [21] is shown, with a remarkable agreement of the TM domains, as suggested by the 1.97 Å RMSD f for the $C_\alpha$ atoms. The hinge α-helix (P272-N280) and the P10 loop (F353-Y362) at the interface between LRR and TM were well preserved in the alignment, including the position of the disulfide bridges C276-C356 and C275-C346. Comparison of the HR region was rather uncertain given that residues I296 to M328 were not resolved by the cryo-EM structure [21]. In previous studies, the FSHR-TM model, including D355 to N678, was developed before the breakthrough of the AF2 method for predicting native structures. In S1 Fig we show a structural alignment of the FSHR-TM model in comparison to the complete AF2 model of FSHR;

**Table 1. Relative distances among the LRR, HR, and TM domains of the receptors' structures and the FSHR AF model.**

| Distance | LHCGR<br>PDB:7II | FSHR<br>AF: P23945 | FHSR*<br>PD:B8I2G |
|---|---|---|---|
| **LRR-HR** | (S55-C304)<br>66 Å | (L31-Y303)<br>63 Å | L(31-S295)<br>48.3 Å |
| **TM-HR** | (L452-C304)<br>32 Å | (L460-Y303)<br>56Å | (L460-S295)<br>76.289.4Å |
| **LRR-TM** | (S55-I431)<br>32 Å | (L31-L460)<br>56Å | (L31-L460)<br>89.4Å |

*Distances of the HR domain were calculated using S295, instead of Y303, due to missing residues in the cryo-EM structure at the HR domain.

both models showed consistent predictions of the TM helices, including, for example, the location of cysteine residues. The good agreement among the FSHR models and the FSHR cryo-EM structures provides an additional criterium to validate predictive models [15].

By structurally aligning the LRR of the FSHR (A48-T270) and the LHCGR (T52-T274), we calculated a RMSD of 2.40 Å; for the HR domain of the FSHR (P276-T345) and the LHCGR (P276-P345), the RMSD was 17.9 Å. By aligning the sequence of the TM of the FSHR (Y362-F630) and the LHCGR (D360-T630), we calculated a RMSD of 6.1 Å; both structures were consistent with the active (bound to G-protein) state conformation. The largest difference between the FSHR and LHCGR was observed in the HR domain, which deserves further detailed analysis because of its role on triggering the activation process of these receptors [20,31]. In this study, we explored the motion of the LRR, HR, and TM domain in a conformation consistent with the active state. For this purpose, we defined positions at each domain and calculated the relative distances (Table 1), which may serve as reaction coordinates to follow the transition among conformational states. For the LRR we chose a residue at the middle of the first β-strand, for the HR a residue at the middle of the α-helix, and for the TM domain a residue at the middle of helix 3.

The comparison among relative distances LRR-TM, LRR-HR, and TM-HR are shown in Fig 2, including the correlation coefficient for each replicate. The overlapping areas represent conformations with the same values of relative distances (internal coordinates) and larger

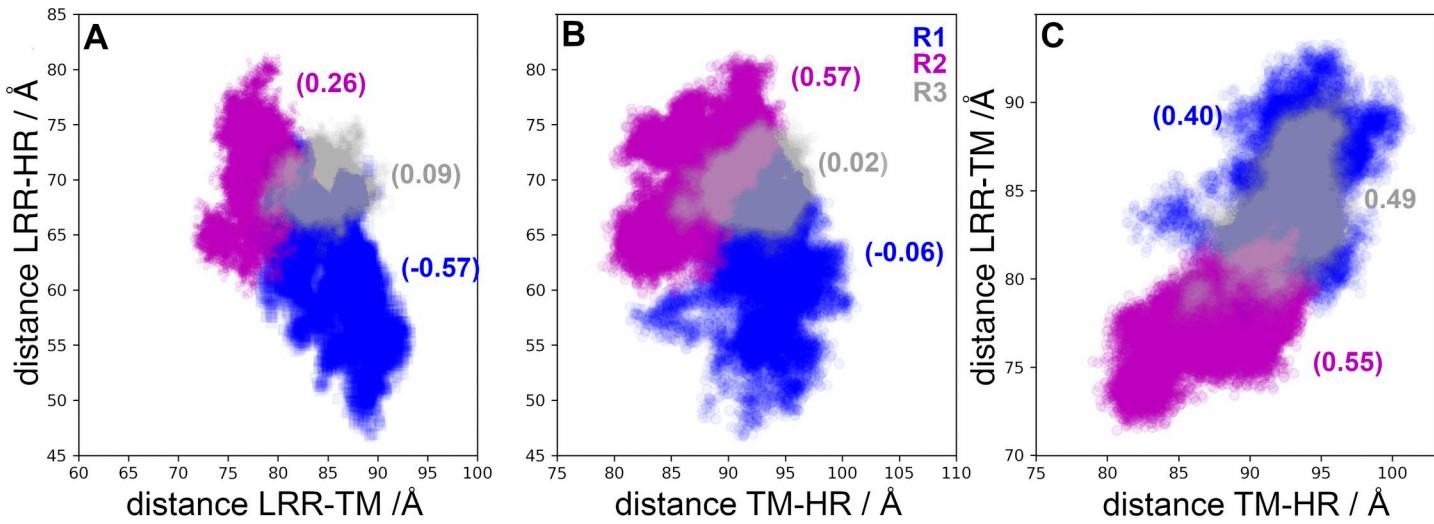

**Fig 2. Distances among structural domains of the FSHR.** (**A**) Distance LRR-HR *vs* LRR -TM. (**B**) Distance LRR-HR *vs* TM-HR. (**C**) Distance LRR-TM *vs* TM-HR. Numbers in parenthesis indicate the correlation coefficients for every pair of distances. Color code: R1-blue, R2-magenta, and R3-gray.

colored areas represent broader distributions. In R1, distributions were mainly unimodal centered at 70 Å, 85 Å, and 93 Å, for LRR-HR, LRR-TM, and TM-HR distances, respectively. Positive correlations calculated for distances LRR-TM and TM-HR were of 0.40, 0.55, and 0.4, for R1, R2, and R3, respectively (Fig 2C). The increase of the LRR and HR distance relative to TM domain was consistent with a shortening of the LRR and HR distance, as can be observed in R1 by the negative correlation between LRR-HR and LRR-TM. From a comparison among the distances of the FSHR models and the cryo-EM structure (Table 1), and the fluctuations observed in the MD trajectories (*eg.* 80 to 100 Å between TM domain and HR, or 72 to 93 Å between LRR and TM domain) it was evident that the sampling of the configurational space populated conformations consistent with the resolved structure. Interesting, in replicate R3 the distance variability was smaller than in replicates R1 and R2, which can be related to the sampling of a metastable state. Metastable states may prevent transitions among intermediaries and alternative strategies must be implemented for an extensive sampling of the conformational space. For example, trajectories can be generated with configurations harvested from a previous run, as it is described below.

The correlations for distances in the LHCGR are shown in Fig 3 for replicates R1-3 and for the trajectory in POPC. From the trajectory in POPC, receptor configurations were harvested at t = 0 ns (R1), t = 100 ns (R2), and t = 180 ns (R3). Given the initial configurations of R2 and R3, these trajectories displayed no overlaps with replicate R1. Instead, the trajectory in POPC connected the explored regions by R2 and R3 with R1. Interesting, the trajectory in POPC showed multimodal and broader distributions than replicates R1-3 for distances LRR-HR and TM-HR (Fig 3B). All trajectories consistently showed negative correlations between LRR-HR *vs* TM-HR (Fig 3B), which indicated that the HR was moving away from the LRR, approaching to the TM domain. Correlations of distances LRR-TM *vs* TM-HR in the LHCGR (Fig 3C) showed different trends than those in the FSHR (Fig 2C); the concerted motion of the LRR and HR relative to the TM domain in FSHR was not detected in the LHCGR. In summary, from the sampling of receptor conformations in independent replicate trajectories, starting from the same initial configuration (with restarted velocities) or from conformations harvested from a previous trajectory, it was possible to prevent sampling of metastable states. In fact, the weighted ensemble (WE) method takes advantage of generating multiple, short trajectories departing from different initial configurations and restarting trajectories, whenever the

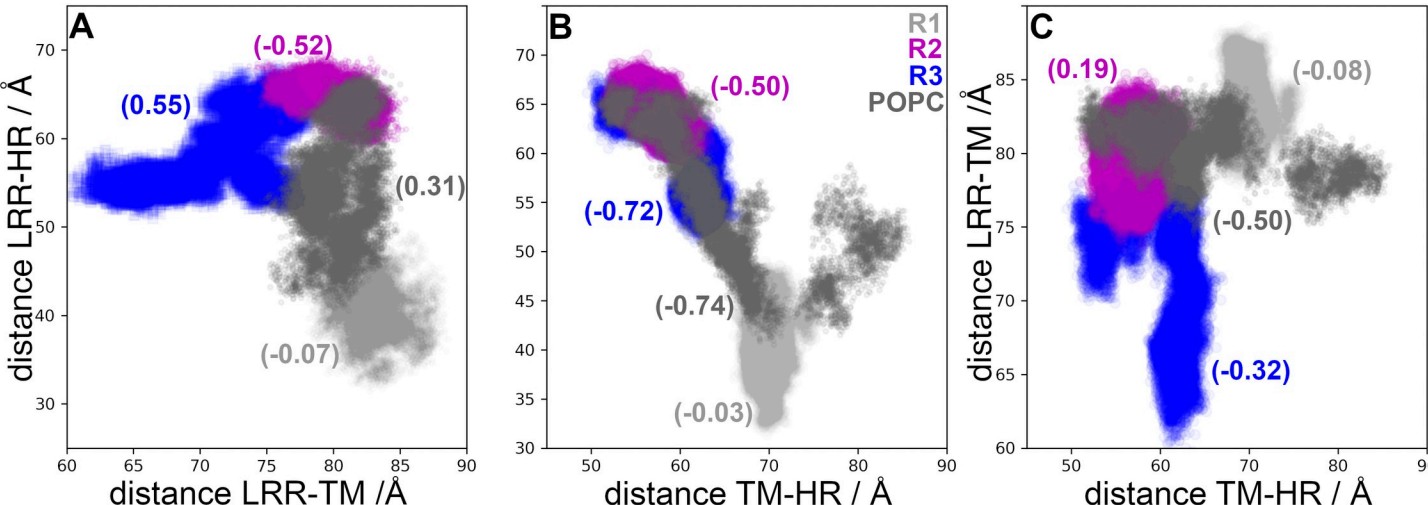

**Fig 3. Distances between structural domains of the LHCGR.** (**A**) Distance LRR-HR *vs* LRR -TM. (**B**) Distance LRR-HR *vs* TM-HR. (**C**) Distance LRR-TM *vs* TM-HR. Numbers in parenthesis are the correlation coefficients for every pair of distances. Color code: R1 gray, R2 magenta, and R3 blue, POPC carbon gray.

reaction coordinate populates a new bin along the state's transition [32]. A plot of the RMSD for the active state of the LHCGR obtained from 100 iterations implementation of the WE method is shown in S5 Fig In fact, the RMSD can be used as a reaction coordinate to track the transition between the active to inactive states.

### Conformational analysis of the LHCGR

Fig 4 shows RMSD matrices for the LHCGR replicates R1 to 3 and the POPC; insets also show cluster conformational analysis with a cut-off criterium RMSD <0.5 Å. For the RMSD

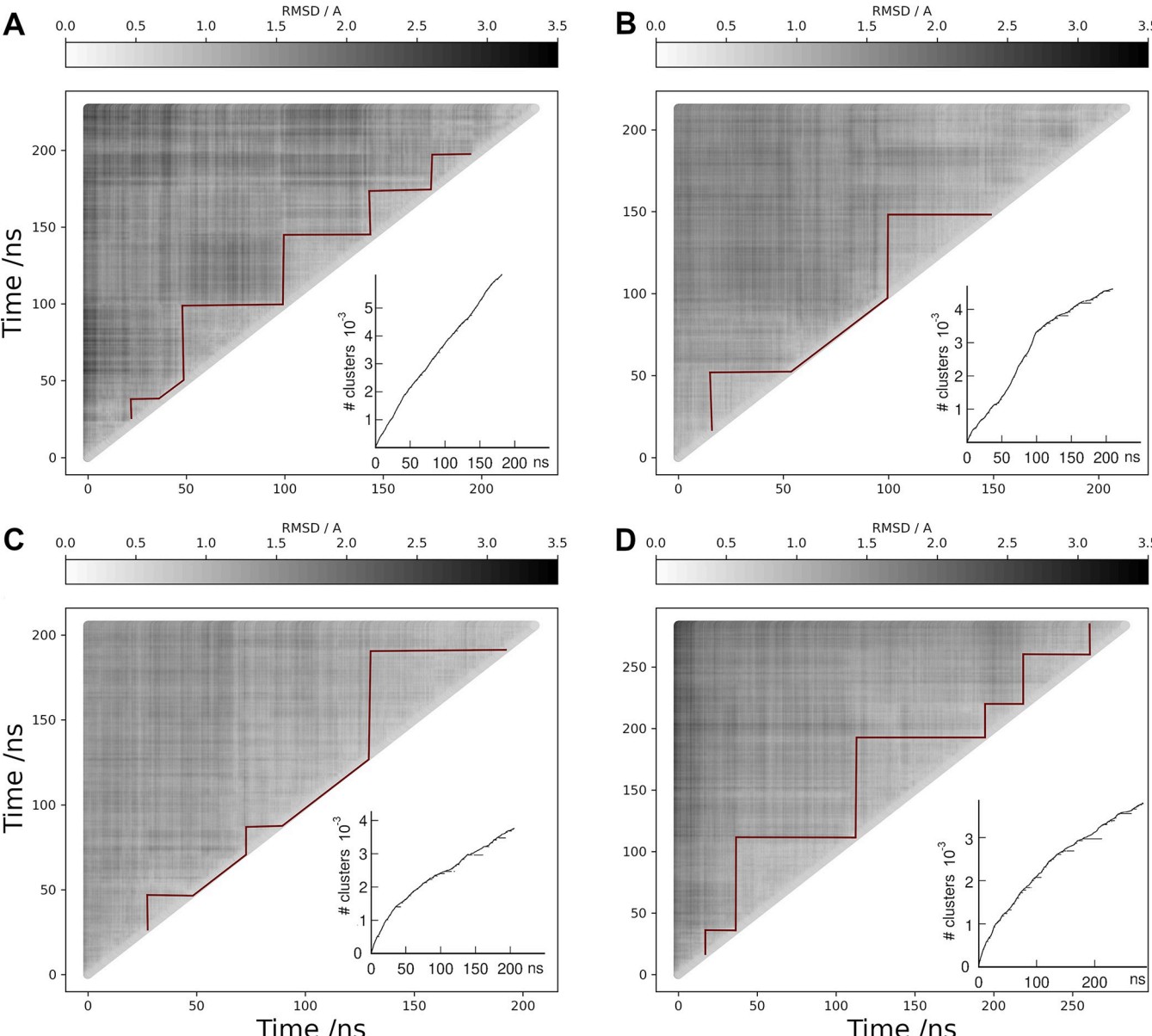

**Fig 4. RMSD matrix analysis (gray scale) for the TM helices of LHCGR.** (**A**) replicate R1, (**B**) replicate R2, (**C**) replicate R3, and (**D**) trajectory in POPC. Red solid lines along the diagonal were drawn for visual identification of self-similar groups. The cluster analysis is shown in the insets. Conformations in clusters include structures within 0.5 Å of RMSD among each other. The cluster number is identified in the vertical axis. Horizontal segments along the curve identify the time frames forming a given cluster.

calculation, only the TM helices were fitted in time frames at *t* and *t+Δt*: TM1, 360–387; TM2, 392–420; TM3, T437-470; TM4, 480–504; TM5, 522–552; TM6, 565–597; and TM7, 602–625. The red solid line in the plots was drawn to identify self-similar groups. Not surprising, similar structures were found along the diagonal as conformations in consecutive frames differed within the cutoff criterium. RMSD values in the TM helices 1–7 were lower than 3.1 Å in SDPC and lower than 2.8 Å in POPC. In Fig 4A, self-similar groups were detected at the ~10 ns time scale in R1. In R2, a group was detected in the first 50 ns and other after 100 ns (Fig 4B). In R3, RMSD values showed closer differences in comparison with R1 and R2. In the cluster analysis, a structure was added whenever it showed a RMSD within the cut-off of any of the members in that cluster. For example, with a cut-off of 1 Å only one cluster was detected; therefore, we set a 0.5 Å for detection of larger number of clusters. Clusters were calculated for each replicate in SDPC and the POPC trajectory, altogether encompassing ~800 ns of conformational sampling.

PC analysis for the LHCGR runs is shown in Fig 5 by means of 2D projections of replicates R1 to 3 over the first four eigenvectors. For the largest clusters found in each replicate, the set of conformations were also projected over the eigenvectors: cluster 6383 of R1, clusters 4193 and 2595 of R2, and clusters 1404 and 2964 of R3. Conformations in clusters were plotted in the context of the full trajectory (Fig 5). Further, in Fig 6 the distributions for each PC of the replicates are shown, providing an additional context in which the clusters were located. For example, conformations in cluster 6383 of R1 were not at the maximum of -8 nm, instead they

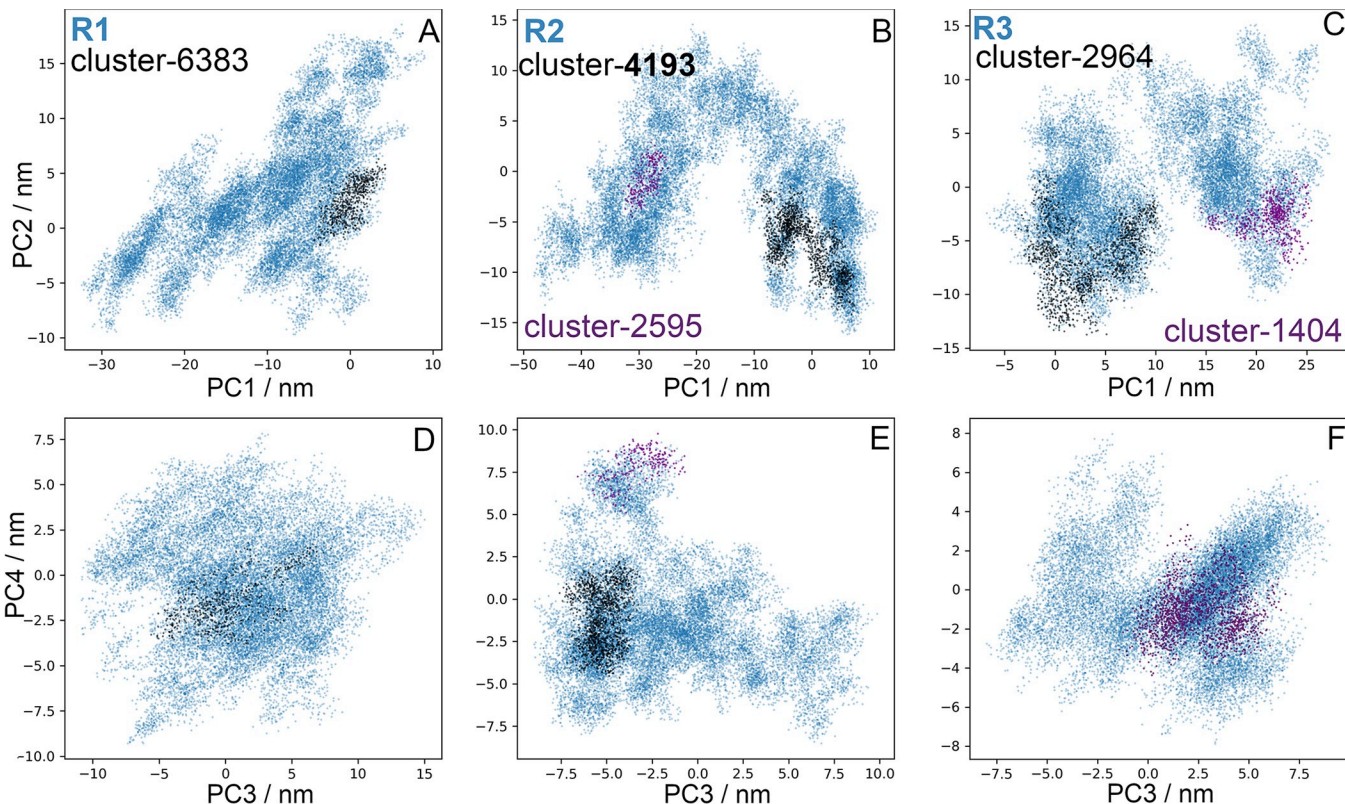

**Fig 5. Principal component analysis for the LHCGR.** PC1 and PC2 (**A-C**) and PC3-PC4 (**D-F**) of trajectories R1-3 (blue dots). Projections for cluster 6383 of R1 (**A, D**), 4193 of R2 (**B,E**) and 2964 of R3 (**C,F**; black dots); clusters 2595 of R2 (**B,E**), and 1404 of R3 (**C,F**; lavender dots). TM domain conformations in clusters were identified with a 0.5 Å RMSD cutoff, *ie*. a structure was included whenever its RMSD was lower than 0.5 Å from any of the members in that cluster.

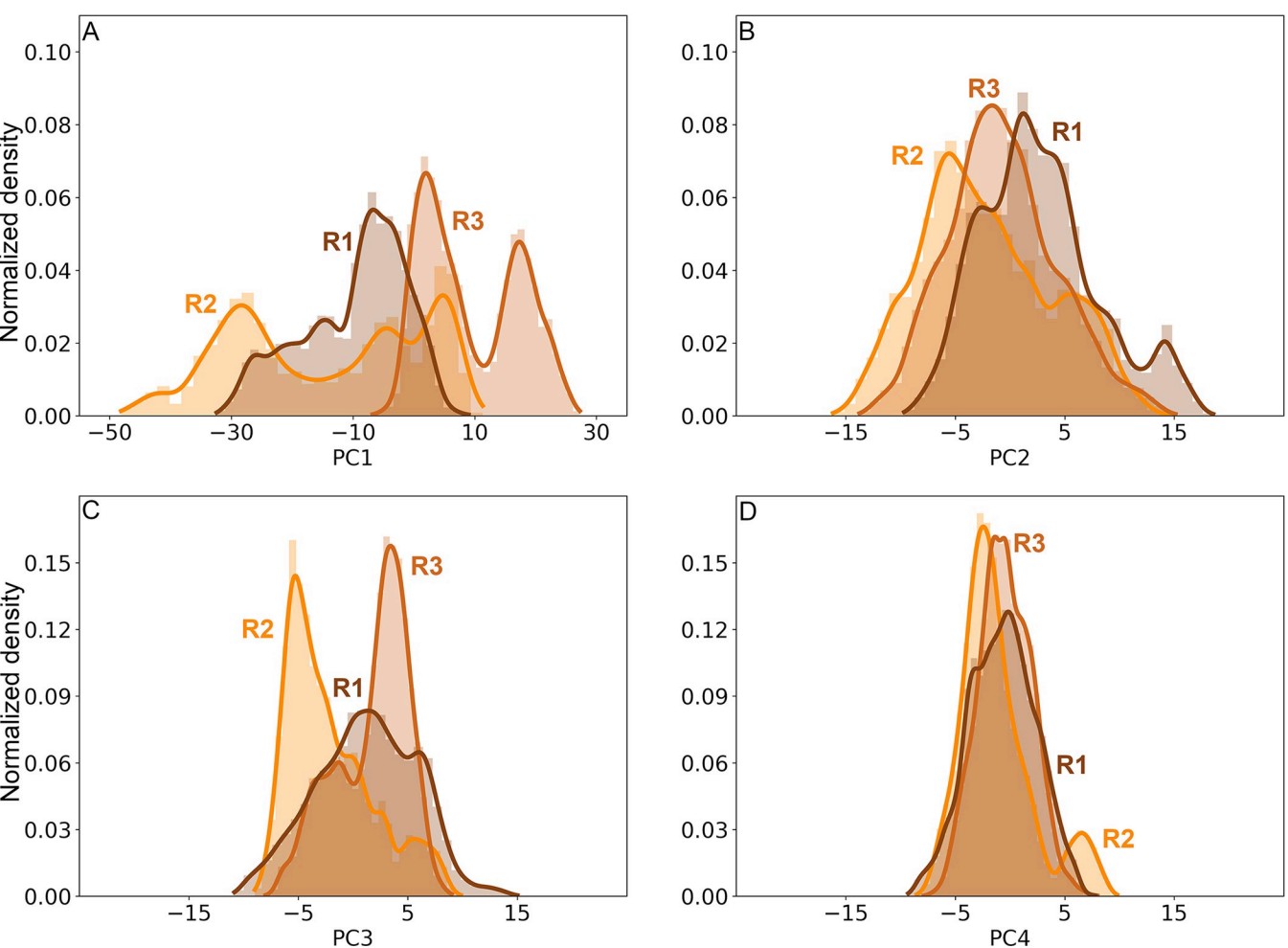

**Fig 6.** Distributions of the trajectory projections on the first four eigenvectors for the LHCGR in replicates R1 (brown), R2 (orange), and R3 (brown-orange). PC1 (**A**), PC2 (**B**), PC3 (**C**), and PC4 (**D**).

were located at 0 nm near the tail of the distribution (Fig 5A). In R2, the PC1 varied from -50 nm to 10 nm, with maxima at -30 nm, -8 nm, and 8 nm; cluster 4193 contributed with conformations at the ±8 nm maxima, whereas cluster 2595 did so with conformations at -30 nm. Cluster 2964 of R3 had conformations around the maximum at 2.3 nm of PC1; cluster 1404, had conformations around the maximum of 22 nm (Figs 5C and 6A). Because of the broad shape of the PC3 of R1 (Fig 6C), cluster 6383 showed values that spread from -5 nm to 5 nm, but right on maximum of PC4 at -2.7 nm (Fig 6D). Cluster 4193 of R2 showed a population at the maxima of PC3 (-5.4 nm) and PC4 (-4.0 nm); and cluster 2595 at 7.5 nm of PC4 (Figs 5E and 6D). Finally, clusters 2964 and 1404 of R3 (Figs 5F and 6C) had conformations at maxima of PC3 (3.8 nm) and PC4 (3.3 nm; Fig 6D). By combining cluster and PC analyses we could identify those conformations that most likely contributed to low frequency motions, that is, the *intermediary states* during conformational transitions in a rough free energy landscape.

## Conformational analysis of the FSHR

The motion of the receptor LRR and HR domains detected by the relative distances can be related to the fluctuations of the TM domain as a correspondence between the dynamics of the

aqueous and membrane environments. It seems evident that by including the variability of the LRR and HR domains together with the fluctuations of the TM helices in a conformational analysis, precludes the identification of the transition states due to the dissimilar nature of the conformational transitions [29]. The RMSD matrices and the cluster analysis for replicates R1-3 of the FSHR are shown in Fig 7. Only the TM helices were fitted for structures taken at time t and t +Δt: TM helix-1, Y362 to Y392; TM helix-2, P397 to Y432; TM helix-3, N437 to T472; TM helix-4, A487 to G507; TM helix-5, M532 to R557; TM helix-6, D567 to L597; and TM helix-7, A607 to Y626. Calculated values of RMSD were lower than 2.5 Å, 2.2 Å and 1.9 Å, for R1, R2, and R3, respectively. The cluster analyses are included in insets of Fig 7, using a 0.5 Å RMSD cutoff; a structure was included whenever RMSD <0.5 Å from any of the members of a given cluster.

The 2D projections of replicates R1-3 on the first four eigenvectors are shown in Fig 8: PC1 *vs* PC2 and PC3 *vs* PC4. The RMSF of PC1 represents ~70% of total fluctuation, whereas the RMSF of PC1 to PC4 ~95% (S6 Fig). Clusters 3405, 1171, and 0733 were identified for R1, R2, and R3, respectively, with the largest number of conformations within the RMSD cutoff. To identify the most populated intermediaries in the context of the full trajectory, projections of the clusters on eigenvectors were included in Fig 8. In addition, Fig 9 shows the distribution of the PC1 to PC4 for each replicate. Cluster 3405 of R1 showed values of -0.5 Å in both PC1 and PC2 (Fig 8A), which was consistent with the maxima of their distributions (Fig 9A and 9B). Due to the broad distributions of PC3 and PC4 (Fig 9C and 9D), conformations in cluster 3405 were as disperse as the full trajectory. Cluster 1171 of R2 showed conformations around the maxima of PC1-2, and PC4 (Fig 8B and 8F), according to its corresponding distributions (Fig 9). Identification of conformations at the maxima of the main PC may underpin an important intermediary. In R3, distribution of PC1 displayed a prominent maximum at -16 nm and a secondary maximum at -25 nm (Fig 9A); PC2 showed a broad distribution with four maxima from -20 nm to 15 nm (Fig 9B). Cluster 0733 of R3 showed a conformation at maxima of all PC, albeit not in the principal maximum of PC3. In summary, the most populated clusters whose conformations of the TM domain exhibited a RMSD <0.5 Å, populated the PC distributions with conformations showing high probabilities. By identifying less populated clusters, the intermediates in values at shoulders or minimum of the PC distributions might be useful to establish possible transition states.

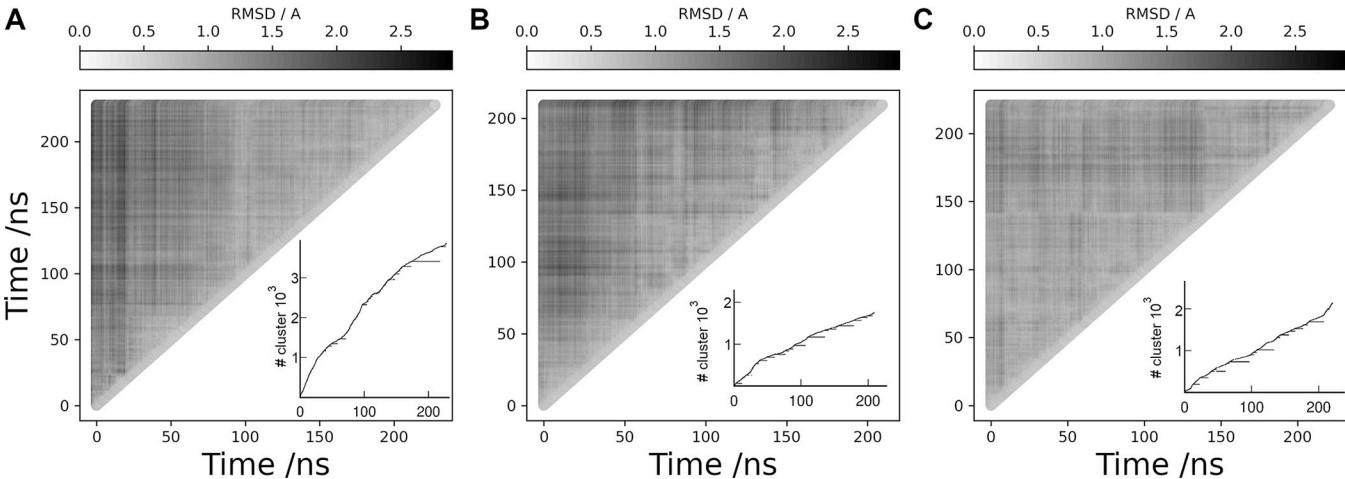

**Fig 7. RMSD matrix analysis (gray scale) for the TM helices of the FSHR. A**. replicate R1; **B**. replicate R2; and **C**. replicate R3. The corresponding cluster analyses are shown in the insets. Conformations in clusters include structures within 0.5 Å of RMSD among each other. The cluster number is identified in the vertical axis. Horizontal segments along the curve identify the time frames forming a given cluster.

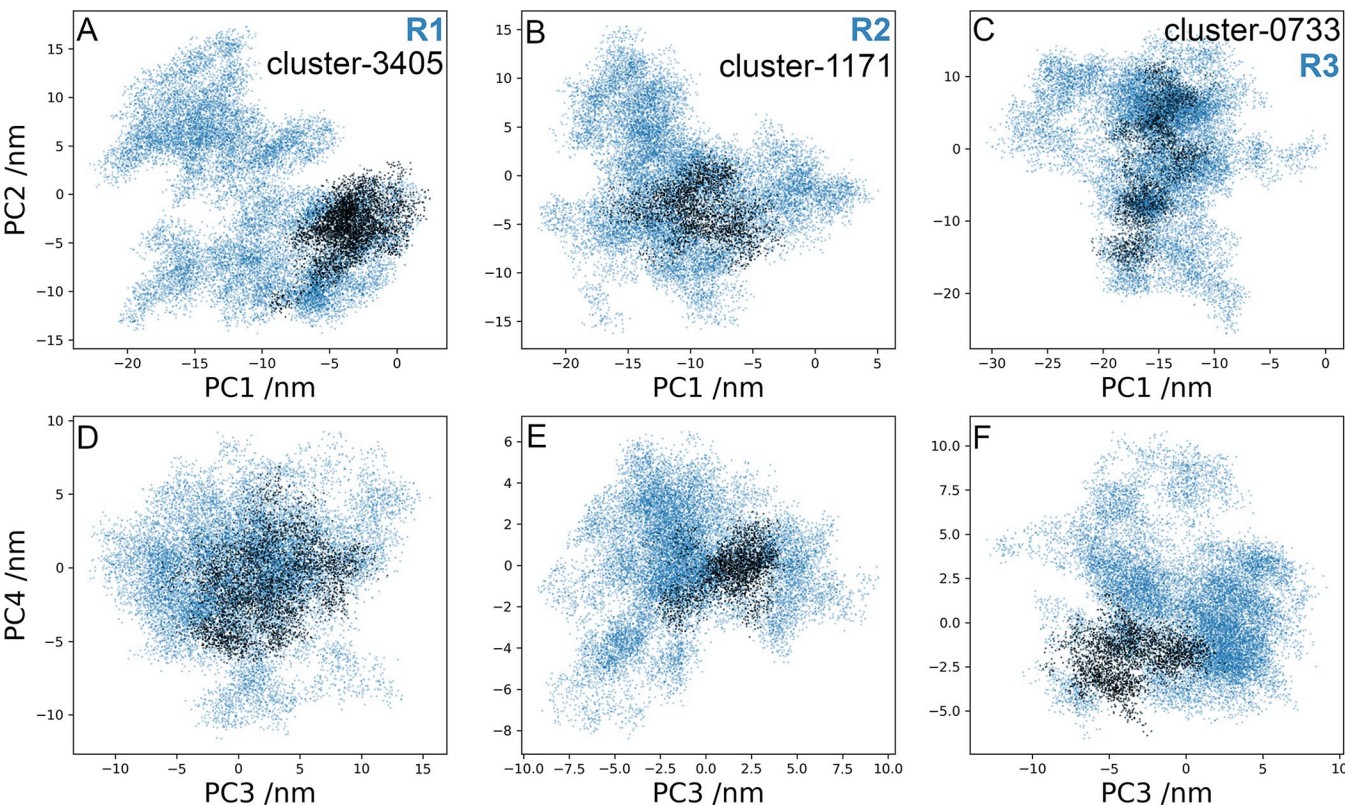

**Fig 8. Principal component (PC) analysis in the FSHR.** Projections on PC1-PC2 (**A-C**) and PC3-PC4 (**D-F**) of trajectories R1-3 (blue dots). Projections for clusters 3405 in R1, 1171 in R2, and 0733 in R3 are included (black dots). Clusters include conformations whose RMSD difference is lower than 0.5 Å in the TM domain region.

A 3D representation of the collective motion related to the PC1 is shown in Fig 10. The arrows at the $C_\alpha$ atoms represent the direction and amplitude of the motion as a result of the projection of trajectories on the first eigenvector. Among other interesting features, in FSHR (Fig 10A) the LRR domain showed a similar motion in the R1 to R3 replicates, moving towards the membrane and consistent to a transition towards the inactive conformation [21]; the LRR motion was accompanied by the HR domain, also towards the membrane, which is consistent with the positive correlations of the distances LRR-TM and TM-HR of Fig 2C; the dynamics of the receptor was mainly dominated for the extracellular regions, as the motion of the TM domains was small in comparison to the motion of the LRR and the HR (S1 Movie). In LHCGR, the LRR motion was divergent in R1-R3 trajectories regarding the directions (Fig 10B). In R1, LRR moved backwards as the HR moved towards the membrane; in R2, LRR tilted towards the membrane, similar to the transition to the inactive state [19] as the HR moved upwards away from the membrane; in R3, LRR moved in opposite direction from R2, similar to transition to the active state, and the HR moved (forward) parallel to the membrane (S2 Movie). Apparently, the LHCGR comparatively showed more dynamics than the FSHR, which could be a signature of the constitutive activation of the gonadotropin hormone receptors [33].

## Correspondence between membrane and aqueous dynamics of the receptors' domains

Clusters conformed by the cutoff criterium (RMSD <0.5 Å) may be related to motions of the extracellular domains, LRR, HR, and TM. Fig 11 shows distances for the clusters identified in

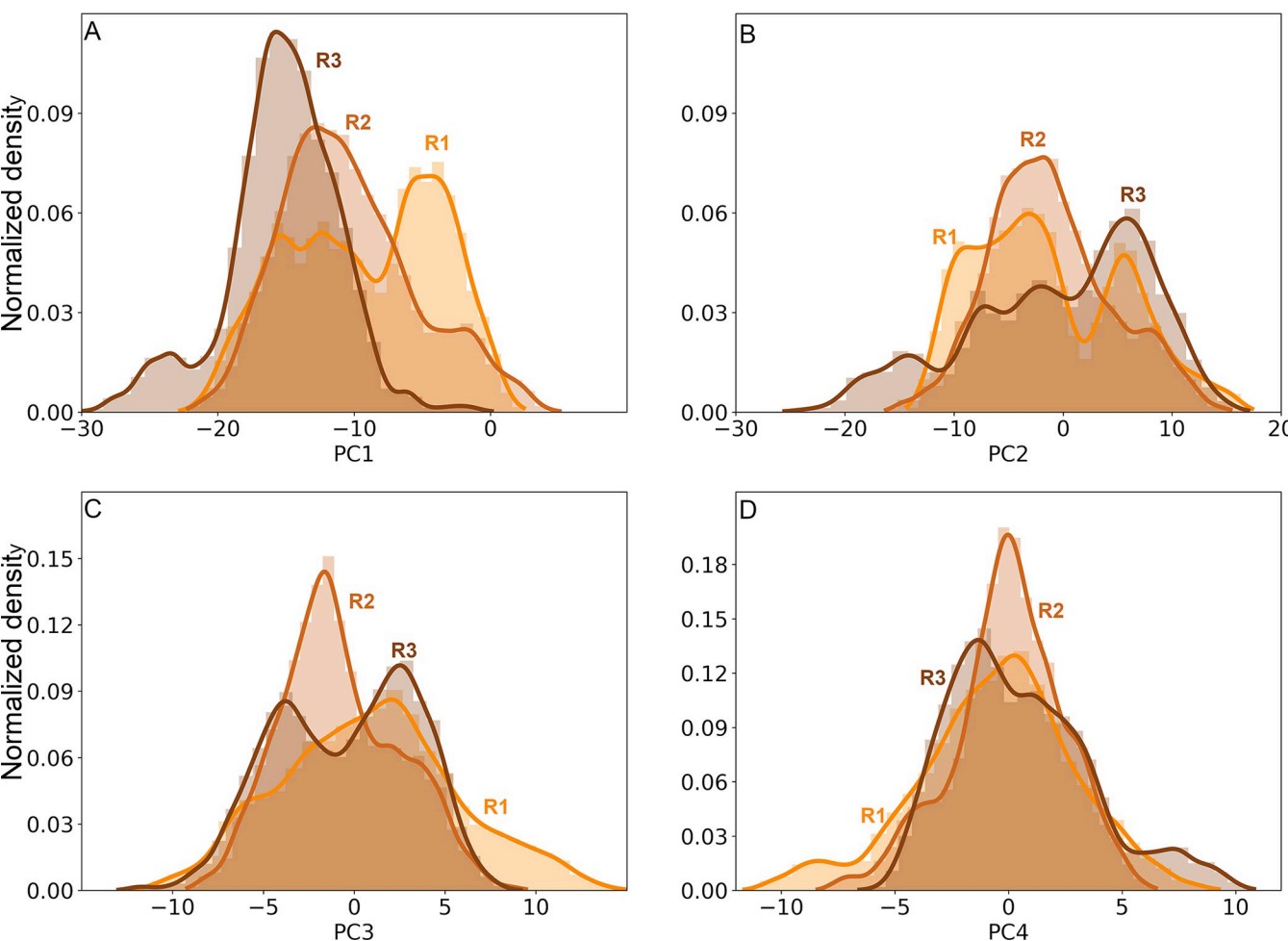

**Fig 9. Distributions of the trajectory projections on the first four eigenvectors for the FSHR in replicates R1 (orange), R2 (brown-orange), and R3 (brown).** PC1 (**A**), PC2 (**B**), PC3 (**C**), and PC4 (**D**).

replicates R1-3 of the LHCGR. Cluster 6383 was detected for R1 (Fig 11, row R1); however, the cutoff criterium yielded too many clusters with few conformations (Fig 4A). To circumvent this problem, the cutoff can be adjusted to values 0.5 <RMSD< 1.0 in order to conform more populated clusters. In fact, for R2 the RMSD cutoff criterium allowed to identify distinct clusters in terms of the relative LRR, HR and TM distances (Fig 11, row R2); hence, each cluster corresponds to a different state of relative distances. In R3, there were also distinguishable clusters, albeit in this case there were overlapping conformations most likely because clusters 2407 and 2964 were close in time (Fig 11, row R3). In addition, in R3 it was possible to detect the negative correlation between RRL-RB *vs* TM-RB (Fig 3B). From the analysis of relative distances in clusters, it was possible to identify conformational states of the LHCGR in MD trajectories for independent replicates, with starting configurations at different times of a previous run. Using both, the RMSD and any of the relative distances, it is possible to follow transitions among conformational states (*eg.* active—inactive) with these reaction coordinates.

Fig 12 shows the relative distances for the FSHR and, in this particular case, for only the most populated cluster of each replicate R1 to 3. Conformational states can be identified in terms of the relative distances. In particular, R2 showed almost no overlapping conformations with R1 and R3. Because the trajectories were independent, the clusters explored different

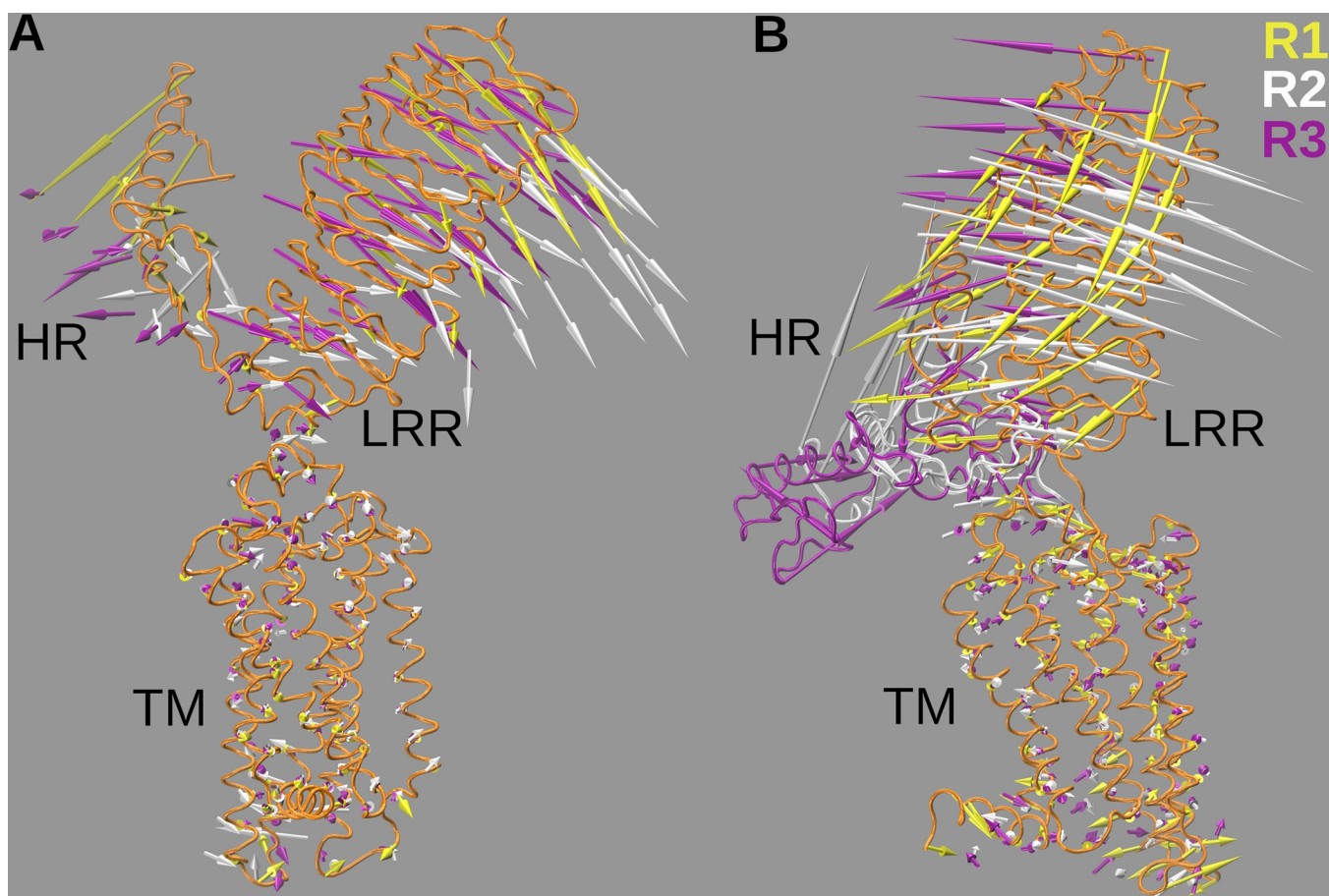

**Fig 10. Projection of the first principal component (PC1) on the 3D structure of the gonadotropin receptors. A.** FSHR (orange structure) with arrows on $C_\alpha$ atoms that represent the amplitude and directions of the motion of PC1. **B**. LHCGR (orange structure) with arrows on $C_\alpha$ atoms that represent the amplitude and directions of the motion of PC1. Color code for arrows: yellow-R1, white-R2, and lavender-R3. Motion of the extracellular regions dominated the dynamics of the receptors. The LHCGR showed divergent motions in replicates R1 to R3 as trajectories started from configurations, respectively, at 0 ns,100 ns, and 180 ns of the POPC run.

regions of the conformational energy landscape. Additional comparisons can be made among replicates; for example, it is possible to project a trajectory over the eigenvectors of a second trajectory. In S7 Fig, a projection of R1 over the first eigenvector of R2 provides information on the conformations of R1 that contributes to PC1 of R2; conversely, a lack of overlap among distributions would represent that trajectories show different dynamics in the conformational space. By the strategy applied in this study, we could distinguish conformations of the FSHR in which the HR moved relative to LRR, as suggested by the previously proposed activation mechanism [7].

## Discussion

One of the key physiological functions of GPCRs is signal transduction, that is, the triggering of a cell response to a particular (agonist) stimulus. Nevertheless, these receptors are not necessarily a switch that turns on and off in response to a stimulus rendering a single or unique intracellular response [18]. Conformational variability might explain that a particular receptor may be coupled to diverse signaling molecules related to different concepts such as allosteric regulation, signal predisposition or selective signaling [34,35]. Given that the structural

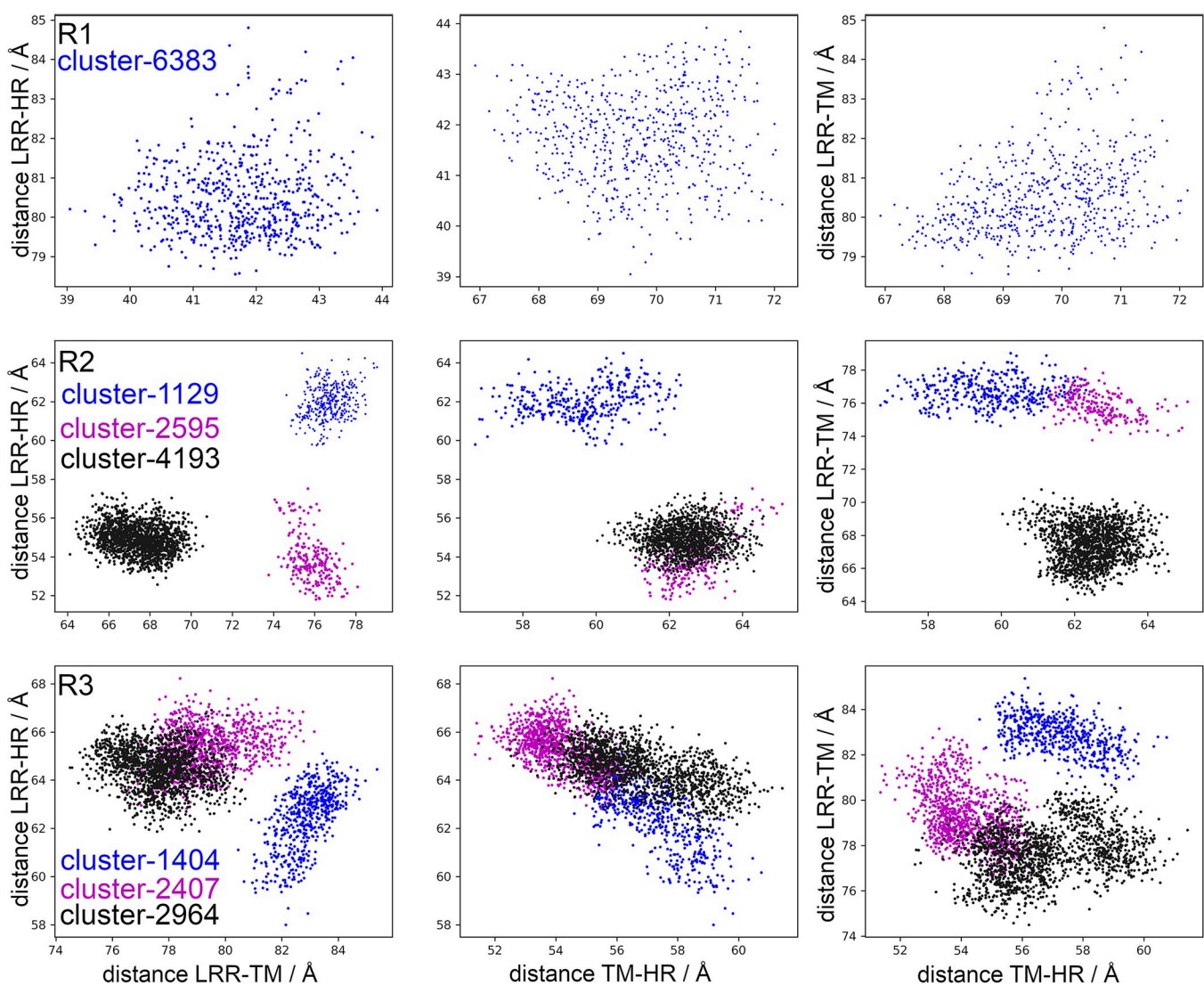

**Fig 11.** Distances among the LRR, HR and TM domain in the LHCGR (Å) in R1 (upper plots), R2 (middle plots) and R3 (lower plots). Analysis for clusters 6383 of R1; 1129, 2596, and 4193 of R2; and 1404, 2407, and 2964 of R3.

information to determine the intermediate states of GPCRs during the activation process is scarce, *in silico* MD techniques have resulted quite useful to establish their structure-function relationship in dissimilar environs, from the phospholipid membrane core to the bulk aqueous medium. In particular, we employed a FSHR model that included the LRR, HR, and TM domains as it appears in the AF2 server [36,37], whereas the LHCGR structure was obtained from the PDB repository (access code 7FII) [19]. Both receptors were analyzed in a comparative manner following the same computational protocols. The internal coordinates were defined for measuring distances among the LRR, HR and TM domains to detect the relative motion of the HR, which is recognized as a key region involved in the activation of all GPHRs [20,31,38].

The HR exhibits an α-helix segment, a P10 segment, and a loop which extends to the aqueous medium and that resembles the thumb of a glove (Fig 1). Based on the structure of the LHCGR, the activation mechanisms proposed consists in the displacement of the LRR to a

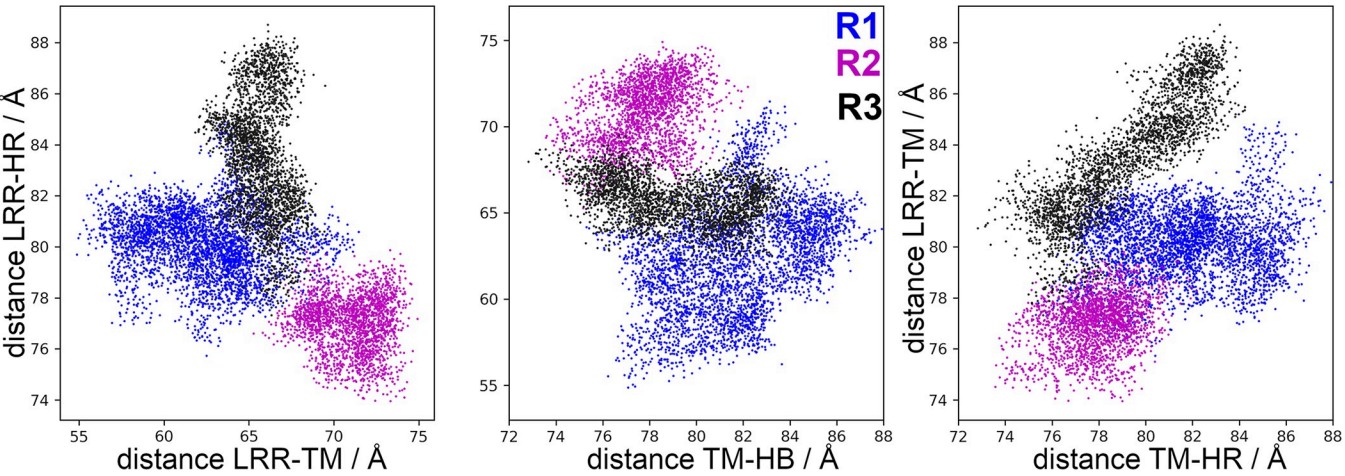

**Fig 12.** Distances among LRR, HR and TM domains of the FSHR (Å). Blue dots, group 3406 of R1; magenta dots, group 1171 of R2; and black dots, group 0733 of R3.

vertical position with respect to the plasma membrane (*ie.* the TM-LRR distance increases), while the HR approximates to the membrane (*ie.* the TM-HR distance decreases) [19]. The present *in silico* MD simulation of the LHCGR, generated trajectories that exhibited a relative motion in which the LRR-TM fluctuated between 59.7 and 88.6 Å, the LRR-HR between 31.4 and 70.0 Å, and the TM-HR between 48.2 and 79.2 Å. Motions of the TM-HR and LRR-HR consistently exhibited negative correlations, where the increase in the former was linked to decrease in the latter (S2 Movie and Fig 10). In the crystal structure of this receptor, the presence of the agonist may prevent the proximity between the LRR and the HR; these extreme values might be useful to implement weighted ensemble simulations [32].

Although the role of the HR as an activation switch in GPCRs like the FSHR has been recently challenged [38], it has been proposed that during FSHR activation a displacement of the HR that allows insertion of residue Y355 in the interface between the α and β subunits of FSH occurs [7,39]. In the active state, the LRR is detected in vertical position (perpendicular to the membrane) and the HR must move to the LRR [19,40]. That is, the HR movement consists in an increase in LRR-TM distance and a corresponding decrease in the LRR-HR one, leading to a negative correlation between those distances. In the FSHR the relative motions of the LRR-TM would fluctuate between 71.4 and 93.3 Å, those of the LRR-HR between 46.7 and 81.3 Å, and those of the TM-HR between 78.9 and 101.2 Å. Hence, trajectories using configurations with increasing TM-LRR and/or decreasing LRR-HR distances, along with fluctuations of RMSD at the TM domain, may be useful to identify transition intermediaries.

Another important element associated to the motion of the HR is the rearrangement of the hinge α-helix (Y271 to W281) and the P10 loop (F353-Y362), both located at the interface between the LRR and the extracellular loops. In the S3 Movie we show the projection of the PC1 on FSHR, with a close up at the P10 loop and the hinge α-helix. The hinge α-helix binds to the LRR and P10 loop by disulfide bonds (Fig 1) important for preserving the integrity of the holo-FSHR [21]. In addition, the hinge α-helix also showed a displacement in the active conformation relative to the inactive one, mainly moving away from the extracellular loop 1, suggesting the role of this region as a bridge for connecting the structural changes of the LRR and TM domain [21]. At the TM domain, binding of agonist was linked to the motion of helix 6, in particular a kink between M585 and D581 that increased upon the activating stimulus [30]. In fact, in our simulations we detected fluctuations in the kink angle at the TM helix 6

relative to its angle in the crystal structures (44 and 37 degrees for the FSH and the LHCG receptors, respectively) (S8 Fig). Kink angle averages showed almost 10 degrees increment in the LHCGR, while averages in the FSHR remained close to the reference value (S1 Table). The relative kink increment in the LHCGR was still comparable to that of the FSHR, as the angle in the crystal structures exhibited a 7 degrees difference between the two receptor structures. The kink angle in TM helix 6 might also be an interesting structural parameter for detecting conformational changes in GPHRs.

In the present study, the extracellular and TM domains were initially studied uncoupled due to differences in the dynamics of the aqueous media and the membrane; thereafter, it was possible to identify TM clusters with specific values of relative positions of the LRR and the HR. Importantly, dissection of the conformational sets using PC analyses by including in the fitting procedure the relatively rigid TM domain, allowed us to map the clusters (with a RMSD < 0.5 Å criterium) and the low frequency motions in terms of the PC1-PC4. In addition, we could detect the contributions of clusters to the PC values by projecting the clusters on a given eigenvector, which was very informative on the conformational populations with impact on the low frequency motions. Therefore, to explore the transitional states between the active—inactive states, we propose that LRR-TM, or TM-HR relative distances may be useful as reaction coordinates along with RMSD calculations, cluster analysis, kink angle at helix 6, and/or orientation of α-hinge helix relative to EL1.

The purpose of exploring the energy landscape of the gonadotropin receptors is, among others, to screen for structural variations that may respond to conformational changes leading to well-known states or alternative states (*eg.* those promoting binding of new drugs, or selective coupling to intracellular transducers). A broader perspective may be included by using the reactions coordinates here explored, to bias the conformational dynamics of the receptors upon binding of distinct allosteric modulators or orthosteric agonists that may potentially favor activation of particular signaling pathways, as required by a particular therapeutic purpose.

## Material and methods

Two simulation boxes containing the receptor, a lipid bilayer, water molecules as solvent, and $Na^+$ or $Cl^-$ ions for charge balance were set up. For the FSHR initial coordinates, we used the AF2 [36,37] structure, generated with the full sequence of the human receptor model (AF-P23945). Post-translational modifications were introduced at C644 and C646 by palmitoylation through thioester bonds [22]. Disulfide bonds between the cysteine side chains of C18-C25, C23-C32, C275-C346, C276-C356, C442-C517, and C229-C338 were defined according to the S-S bond distance criterium. Protonation states of side chains were set to those of the predominant species at neutral pH. The FSHR structure was inserted in a preequilibrated bilayer of 1-stearoyl-2-docosahexaenoyl-sn-glycero-3-phosphocholine (SDPC). Water molecules for solvation of the receptor and lipid heads were added in a rectangular box of 120x120x160 Å$^3$, respectively, in the x, y and z directions. A total of 191601 atoms were included: 48411 water molecules, 281 lipids, 3 $Na^+$ ions (3 mM) and the FSHR with 695 residues. The second simulation box contained 190072 atoms: 48414 water molecules, 247 SDPC lipids, 7 $Cl^-$ ions for neutralizing the preserved three Na+ ions (8 mM), and the LHCGR with 613 residues. The LHCGR structure (PDB:7FII) corresponds to the state bound to the hormone and the $G_s$-protein [19], and it was processed according to the CHARMM-GUI membrane builder [41,42] using the following parameters [43]. The chain R (segment PROD) corresponding to the LHCGR receptor's coordinates in the pdb file was selected to be inserted in a POPC lipid bilayer, with box dimensions of 100x100x168 Å$^3$. For the missing residues

T287 to W329 of the HR, the CHARMM-GUI modeling scheme included the coordinates as predicted by the GalaxyFill algorithm [44]. Disulfide bonds were defined between cysteines C279-C343, C280-C353, C131-C156 and C439-C514. The LHCGR principal axis was aligned in the *z* direction and displaced until the TM domain matched the hydrophobic core of the bilayer. The system size was 191601 atoms: 248 1-palmitoyl-2-oleoyl-sn-glycero-3-phospho-choline (POPC) lipids, 37479 water molecules, $Na^+Cl^-$ ions (0.15 M), and the LHCGR. From the assembly of the initial configurations, the systems were energy minimized with conjugated gradient algorithm for 10 k steps, followed by a gradual relaxation of the receptor atoms. In a first stage, the receptor backbone atoms were fixed, then subsequent stages with positional constrains of 20, 15, 10, 5, 3, 2, and 1 kcal /mol $Å^2$, in short trajectories of 200 ps each were applied. The trajectory of LHCGR in POCP was extended for 200 ns without constraints. Because of our interest in exploring the receptor's conformational landscapes, we generated independent trajectory replicates for both the FSHR and the LHCGR in SDPC. In the case of the FSHR, three replicates were generated using the last configuration after the relaxation procedure and restarted velocities for each replicate. For the LHCGR replicates, we used configurations taken from the trajectory in POCP at times 0 ns, 100 ns, and 180 ns, and relaxed the receptor in the membrane environment of SDPC lipids. By changing the lipid environment, from POPC to SDPC, we consider the possibility of enhancement the dynamics of the TM domain in poly-unsaturated lipid tails, as reported on the influence of DHA lipids in the rhodopsin functions [45,46].

All simulations were performed with the NAMD 2.14 software [47], version NAMD3.0 alpha, which was optimized for GPU-accelerated servers [48]. Simulation trajectories were generated in the isothermal-isobaric ensemble (NPT) with Langevin dynamics to maintain a constant temperature [47], and Nosé-Hoover Langevin piston to maintain a constant pressure of 1 bar [49]. Anisotropic cell fluctuations in the x-, y- and z-axis were allowed [50]. Non-bonding interactions were calculated with a cutoff of 11 Å, and a shifting function starting at 10.0 Å. A multiple time step integration for solving the motion equations was used with one step for bonding interaction and short-range nonbonding interactions, and two steps for electrostatic forces, with 2 fs time step. All hydrogen atoms were fixed using the SHAKE and RATLLE algorithms [51,52]. Electrostatic interactions were evaluated using PME [53], with a $4^{th}$ order interpolation on a grid of ~1 Å in the *x*-, *y*- and *z*-directions, and a tolerance of $10^{-6}$ for the direct evaluation of the real part of the Ewald sum. CHARMM36 all-atom force field parameters were used for the lipid molecules [54], and CHARMM36m for the protein atoms [55,56], including CMAP correction [57,58]. Water molecules were modeled using the TIP3P potential [59].

Trajectory analysis were performed in the TCL environment of VMD [60] for the calculation of the root mean square deviation (RMSD), mass density distribution of membrane atoms, and LRR-HR-TM domain distances, among other structural and dynamical parameters. For the principal component analysis (PCA), GROMACS 2020 [61] was employed with commands *gmx covar* for the calculation of covariance matrix excluding the first 20 ns of trajectory, and *gmx anaeig* for analysis of eigenvectors. Trajectory files with only the receptors' backbone atoms (available at https://doi.org/10.5281/zenodo.10011977) were extracted from the full-atom time frames that stored coordinates every 10 ps. We defined a TM group for fitting the trajectory time frames: for FSHR, TM helix-1, Y362 to Y392; TM helix-2, P397 to Y432; TM helix-3, N437 to T472; TM helix-4, A487 to G507; TM helix-5, M532 to R557; TM helix-6, D567 to L597; and TM helix-7, A607 to Y626; and for LHCGR, TM helix-1, 360–387; TM helix-2 392–420; TM helix-3, T437-470; TM helix-4, 480–504; TM helix-5, 522–552; TM helix-6, 565–597; and TM helix-7, 602–625. For the first four eigenvectors we calculated the principal components (PC) for $C_\alpha$ atoms of the entire structure. Calculation of distributions

for PC values (eg. PC1-PC4) were very informative to detect the population of conformational states; for example, those with the higher probability represent a minimum of free energy landscape, and those with lower probability represent transition states [62,63]. To compare conformational populations between replicates, we calculated the distribution of a PC of a second trajectory using eigenvectors of a previous trajectory. Cluster analysis also was performed in GROMACS 2020 [61] with commands *gmx cluster* and *gxm extarct-clust*er. RMSD matrices for comparison of frames at *t* and *t+Δt* were calculated for the same TM group as defined in PCA. Correlations among domain distances also were calculated to detect concerted motions. Visual molecular dynamics (VMD) was used for visualization, representation of 3D structures and image generation [60].

## Supporting information

**S1 Fig. Structural alignment between FSHR-AF2 (cyan structure) and the FSHR-TM (orange structure); for reference, Cα of cysteine residues are depicted as spheres in yellow and in blue for the FSHR-TM and FSHR-AF2, respectively. The open intracellular conformation suggests that the models corresponds to the activated conformation.**
(TIF)

**S2 Fig. Stability of the membrane bilayer in replicates R1-R3 for the FSHR and LHCGR systems.** Stable averages were calculated after 50 nanoseconds for box area (xy-dimensions) and box height (z-dimension), both parameters important for monitoring the equilibration of the membrane bilayer according to the area per lipid and hydrophobic height in SDPC lipid molecules.
(TIF)

**S3 Fig. Mass distribution as function of the bilayer normal.** Water and SDPC functional groups were identified in both, FSHR (top panel) and LHCGR (bottom panel) systems. Calculation were performed for the last 40 ns of trajectory for replicates R1-R3. Distributions are normalized to the number of atoms/$Å^3$.
(TIF)

**S4 Fig.** Root mean squared displacement (RMSD) calculations for the $C_\alpha$ atoms of the transmembrane domains in FSHR (top panel) and LHCGR (bottom panel). Results for R1-R3 replicates were included.
(TIF)

**S5 Fig. Weighted ensemble simulation for the LHCGR (active state) in POPC, with the RMSD as reaction coordinate in Å (dimension 0).** The inactive state was the reference for the RMSD calculation (TM domain). Variability of the RMSD over 100 iterations of 200 ps trajectories populated bins in the interval 3.6 to 4.9 Å. Up to 14 simultaneous trajectories per bin were generated in the simulation to sample states with low probability.
(TIF)

**S6 Fig. Eigenvalues for 15 eigenvectors for replicates R1-3 of FSHR.** The total fluctuation corresponds to the sum of eigenvalues. Contribution of eigenvalues 1 to 4 accounts for the ~95% of root mean squared fluctuation.
(TIF)

**S7 Fig. Distribution for the PC1 of LHCGR in replicate R2.** For comparison, the trajectory of R1 was projected over the first eigenvector of R2. The overlap of distributions represents conformations with similar motion (fluctuations) in both trajectories, otherwise

conformations display different dynamics due to complementary sampling in independent replicates.
(TIF)

**S8 Fig. Kink angle of TM helix 6 relative to the values in the crystal structures.** A. Calculation for FSHR using positions of $C_a$ atoms of L597 and M858 for the upper half, D581 and S565 for the lower half, and a reference value of 44 deg. B. Calculation for LHCGR using $C_a$ atoms of F594 and S586 for the upper half, D578 and C563 for the lower half, and reference value of 37 deg.
(TIF)

**S9 Fig. Root mean squared displacement (RMSD) calculations for the $C_\alpha$ atoms of the transmembrane domains in the LHCGR in POPC. Gray areas represent the interval in which 95% of RMSD values distribute in the R1-R3 in SDPC.**
(TIFF)

**S1 Movie. Projection of the PC1 on the 3D FSHR (orange structure).** Color code: R1-yellow, R2-white, R3-purple. The HR domain for each replicate is shown for visualization of the amplitude and direction of the motion.
(MPG)

**S2 Movie. Projection of the PC1 on the 3D LHCGR (orange structure).** Color code: R1-yellow, R2-white, R3-purple. The HR domain for each replicate is shown for visualization of the amplitde and direction of the motion.
(MPG)

**S3 Movie. Projection of the PC1 on the 3D of the P10 loop and the hinge α-helix in FSHR (orange structure).** Color code: R1-yellow, R2-white, R3-purple. The HR domain for each replicate is shown for visualization of the amplitde and direction of the motion.
(MPG)

**S1 Table. Average of kink angles at TM helix 6.** Calculation for the last 164 ns of the R1-R3 trajectories in FSHR and LHCGR. Standard deviations calculated employing the blocking average method.
(DOCX)

## Author Contributions

**Conceptualization:** Eduardo Jardón-Valadez.

**Data curation:** Eduardo Jardón-Valadez, Alfredo Ulloa-Aguirre.

**Formal analysis:** Eduardo Jardón-Valadez.

**Funding acquisition:** Alfredo Ulloa-Aguirre.

**Investigation:** Eduardo Jardón-Valadez, Alfredo Ulloa-Aguirre.

**Methodology:** Eduardo Jardón-Valadez.

**Project administration:** Alfredo Ulloa-Aguirre.

**Resources:** Eduardo Jardón-Valadez.

**Software:** Eduardo Jardón-Valadez.

**Supervision:** Alfredo Ulloa-Aguirre.

**Validation:** Eduardo Jardón-Valadez.

**Visualization:** Eduardo Jardón-Valadez.

**Writing – original draft:** Eduardo Jardón-Valadez.

**Writing – review & editing:** Eduardo Jardón-Valadez, Alfredo Ulloa-Aguirre.

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
