## [Decision Letter · Decision Letter 0]

18 Sep 2023

Dear Dr Jardón-Valadez,

Thank you very much for submitting your manuscript "Searching deeply into the conformational space of glycoprotein hormone receptors. Molecular dynamics of the human follitropin and lutropin receptors within a bilayer of (SDPC) poly-unsaturated lipids." for consideration at PLOS Computational Biology.

As with all papers reviewed by the journal, your manuscript was reviewed by members of the editorial board and by several independent reviewers. In light of the reviews (below this email), we would like to invite the resubmission of a significantly-revised version that takes into account the reviewers' comments.

Note that a major concern is the convergence of the MD simulations. Addressing this issue will require significant extension of the simulations and/or use of enhanced sampling technologies.

We cannot make any decision about publication until we have seen the revised manuscript and your response to the reviewers' comments. Your revised manuscript is also likely to be sent to reviewers for further evaluation.

Sincerely,

Alexander MacKerell

Academic Editor

PLOS Computational Biology

Daniel Beard

Section Editor

PLOS Computational Biology

Reviewer's Responses to Questions

**Comments to the Authors:**

Reviewer #1: In this manuscript the authors present the results of a set of relatively short (200ns) uncorrelated MD simulations of large complex and highly flexible glycoprotein hormone receptors with the aim of understanding functional conformational transitions that can inform towards allosteric control mechanisms upon ligand binding.

I find the subject interesting, but unfortunately no evidence is provided to support that either system, which were started from a low 4.8 Å resolution cryo-EM (or no resolution, in case of the AF) conformations, can be considered at or near conformational equilibrium. The computational sampling used in the work is limited in its sampling potentials and for that reason not likely to bring such large and complex systems near conformational equilibrium. This is important as the system should be conformationally stable before data production and collection, for the results to be statistically representative and the insight functionally relevant.

As a potentially interesting point, the data obtained for the FSHR can be compared to the cryo-EM structure 8I2G obtained at 2.8 Å resolution to test if the AF structure refinement combined with MD can bring the system to a high-quality structure. Note that the HR domain in the AF structure used in the work is predicted at very low to low confidence level.

As a final point, in both structures the LRR is glycosylated and the root of the N-glycans is resolved in both cryo-EM structures. These glycans should be reconstructed, both for completeness (both hormone receptors are glycoproteins) and because they may very well be functional, and thus useful to the general aims of the work.

There are typos throughout that can be corrected with a careful reading of the draft.

Reviewer #2: This study was performed to identify the conformational changes in FSHR and LHCGR using MD simulations. In these receptors, distances among the LRR, HR and TM domains were measured to determine a key region involved in the activation of the receptor. Finally, they have proposed that LRR-TM, or TM-HR relative distances may be useful to explore the transitional states between the active/ inactive states.

The proposed approach using MD simulation study to understand states between active and inactive seems a strategy with great potential. However, there are several substantial concerns that require the authors' attention.

1. The introduction section contains excessive information regarding PC analysis. In my opinion, it is unnecessary to provide a detailed description of PCs in this section; instead, it would be more appropriate to emphasize their significance within this study and include them as considered parameters in the methods section.

2. The authors have thoroughly discussed the displacement and rearrangement of helices 5-7 in GPCR. However, similar aspects regarding FSHR activation have not been discussed, despite their elucidation in a recent publication (https://doi.org/10.1016/j.compbiomed.2023.106588).

3. In material and methods section, why were both Na+ and Cl- ions used in the second simulation box? It is not mentioned whether they were used to neutralize the system or to provide a salt concentration.

4. The statement is written as “The chain 612 R (segment PROD) of the pdb file was selected to be inserted in a POPC lipid bilayer”. Why was this performed, its not clear in the statement.

5. In line 612 statement, "The chain 612 R (segment PROD) of the pdb file was selected to be inserted in a POPC lipid bilayer," lacks clarity regarding the reason for this action.

6. In the materials and methods section, two different types of systems are described only for LHCGR, but the importance of these two considered systems is not adequately explained.

7. It is not mentioned whether FSHR/LHCGR was simulated with or without a cognate ligand. Without this information, it is unclear how the transition between active and inactive states was expected if FSHR/LHCGR was simulated without a cognate ligand.

8. In this manuscript, the results are exclusively presented in the form of changes in distances among LRR, HR, and TM. It is also crucial to depict the structural changes occurring in these regions for both receptors. It would be interesting if the authors could visually represent the movements and shifts in these domains and then compare the domain movements between FSHR and LHCGR.

9. Authors have discussed that effective motion of the ECD was observed which is important for the activation of receptors. From the manuscript, it appears that this study did not consider agonist-complexed FSHR/LHCGR. If that is the case, how can we expect movement in the ECD to correspond to receptor activation?

A few minor suggestions:

1. The title should be concise.

2. Replace THS with TSHR in the abstract.

3. The abbreviations used in Table 1 appear to be incorrect.

4. In line 573, please verify the acronym "EC2" and replace it with the correct one.

5. The manuscript needs to be carefully reviewed for grammatical errors.

**Have the authors made all data and (if applicable) computational code underlying the findings in their manuscript fully available?**

Reviewer #1: **No: **I haven't seen any electronic data in addition to the manuscript.

Reviewer #2: Yes

PLOS authors have the option to publish the peer review history of their article (what does this mean?). If published, this will include your full peer review and any attached files.

Reviewer #1: No

Reviewer #2: **Yes: **Chandan Kumar
---

## [Decision Letter · Decision Letter 1]

13 Nov 2023

Dear Professor Ulloa-Aguirre,

Thank you very much for submitting your manuscript "Tracking conformational transitions of the gonadotropin hormone receptors in a bilayer of (SDPC) poly-unsaturated lipids from all-atom molecular dynamics simulations." for consideration at PLOS Computational Biology. As with all papers reviewed by the journal, your manuscript was reviewed by members of the editorial board and by several independent reviewers. The reviewers appreciated the attention to an important topic. Based on the reviews, we are likely to accept this manuscript for publication, providing that you modify the manuscript according to the review recommendations.

Sincerely,

Alexander MacKerell

Academic Editor

PLOS Computational Biology

Daniel Beard

Section Editor

PLOS Computational Biology

Reviewer's Responses to Questions

**Comments to the Authors:**

Reviewer #2: The revised manuscript shows improvement with the implemented modifications, and almost all concerns have been addressed. While the manuscript is now more comprehensible, certain issues still require resolution.

1. Authors have calculated the kink at helix 6 (H6) in both FSHR and LHCGR and discussed the importance of H6 fluctuations for receptor activation. It appears from Figure S11 that the angle at H6 of FSHR increased, while in LHCGR, the angle decreased over the simulation time period. However, the observed differences for these two receptors are not discussed in the manuscript.

2. In my previous comment, number 3, the statement regarding neutralizing and salt concentration for the simulation systems is now justified through the response. However, the statement in the manuscript is not very clear.

3. Now it is very clear that both the receptors were simulated without their cognate ligands. However, some statements in the text may be confusing (eg. Line 55-59 in revised manuscript), and it needs clarification that the receptors were not simulated with cognate ligands. Instead, the study focused on exploring various receptor conformations through random thermal motion.

**Have the authors made all data and (if applicable) computational code underlying the findings in their manuscript fully available?**

Reviewer #2: **No: **The structure file for both the receptors and considered parameters for MD simulations can be shared.

PLOS authors have the option to publish the peer review history of their article (what does this mean?). If published, this will include your full peer review and any attached files.

Reviewer #2: **Yes: **Chandan Kumar

Figure Files:

Data Requirements:

Reproducibility:

References:

---

## [Editor Report · Decision Letter 2]

15 Dec 2023

Dear Professor Ulloa-Aguirre,

We are pleased to inform you that your manuscript 'Tracking conformational transitions of the gonadotropin hormone receptors in a bilayer of (SDPC) poly-unsaturated lipids from all-atom molecular dynamics simulations.' has been provisionally accepted for publication in PLOS Computational Biology.

Best regards,

Alexander MacKerell

Academic Editor

PLOS Computational Biology

Daniel Beard

Section Editor

PLOS Computational Biology

---

## [Editor Report · Acceptance letter]

4 Jan 2024

PCOMPBIOL-D-23-01266R2 

Tracking conformational transitions of the gonadotropin hormone receptors in a bilayer of (SDPC) poly-unsaturated lipids from all-atom molecular dynamics simulations.

Dear Dr Ulloa-Aguirre,

I am pleased to inform you that your manuscript has been formally accepted for publication in PLOS Computational Biology. Your manuscript is now with our production department and you will be notified of the publication date in due course.

With kind regards,

Lilla Horvath
